# Coupled Mamba: Enhanced Multi-modal Fusion with Coupled State Space Model

**Wenbing Li    Hang Zhou    Junqing Yu    Zikai Song**[†]  **Wei Yang**[†]

Huazhong University of Science and Technology

{wenbingli, henrryzh, yjqing, skyesong, weiyangcs}@hust.edu.cn

## Abstract

The essence of multi-modal fusion lies in exploiting the complementary information inherent in diverse modalities. However, prevalent fusion methods rely on traditional neural architectures and are inadequately equipped to capture the dynamics of interactions across modalities, particularly in presence of complex intra- and inter-modality correlations. Recent advancements in State Space Models (SSMs), notably exemplified by the Mamba model, have emerged as promising contenders. Particularly, its state evolving process implies stronger modality fusion paradigm, making multi-modal fusion on SSMs an appealing direction. However, fusing multiple modalities is challenging for SSMs due to its hardware-aware parallelism designs. To this end, this paper proposes the Coupled SSM model, for coupling state chains of multiple modalities while maintaining independence of intra-modality state processes. Specifically, in our coupled scheme, we devise an inter-modal hidden states transition scheme, in which the current state is dependent on the states of its own chain and that of the neighbouring chains at the previous time-step. To fully comply with the hardware-aware parallelism, we devise an expedite coupled state transition scheme and derive its corresponding global convolution kernel for parallelism. Extensive experiments on CMU-MOSEI, CH-SIMS, CH-SIMSV2, BRCA, MM-IMDB through multi-domain input verify the effectiveness of our model compared to current state-of-the-art methods, improved F1-Score by 0.4%, 0.9%, and 2.3% on the CMU-MOSEI, CH-SIMS and CH-SIMSV2 datasetes respectively, 49% faster inference and 83.7% GPU memory save. The results demonstrate that Coupled Mamba model is capable of enhanced multi-modal fusion.

## 1   Introduction

Real-world data captured and processed across multiple modalities, such as text, image, video, and sensor data, yield a rich tapestry of information that is inherently complementary. This complementarity profoundly enhances the capacities of deep learning models, facilitating more nuanced interpretations and predictions. As a result, deep learning models that integrate multi-modal data have shown substantial superiority over their uni-modal counterparts in various domains, including visual-language learning [1, 2, 3], multi-modal classification/segmentation [4, 5, 6, 7], sentiment analysis [8, 9, 10, 11] and etc. Given these advantages, the development of effective multi-modal fusion techniques has emerged as a center of attention. A variety of works have explored this topic on convolution or Transformer -based models, and developed specified mechanisms as early, middle, and late fusion, depending on position of fusion been conducted. A more prevalent practice is to first extract features using modality-specific backbones and then devise a fusion module to exploit the complementary information from all modalities. Existing fusion paradigms either aggregate

---

[†]Indicates co-corresponding author.

38th Conference on Neural Information Processing Systems (NeurIPS 2024).

modal-specific features into one by neglecting individual intra-modal propagation [12] or align modal-specific features into a united representation space through regulation while failing to exploit complementary inter-modal information exchange for difficulty in alignment supervision [13].

Recently, the state space models, advanced by the LSSL [14, 15, 16], S4 [17], GSS [18], and S4D [19, 20], use state variables to explicitly model the sequential evolving neural states, have being emerged as compelling alternatives to Transformers for its efficiency in modeling long-range sequences [21]. Particularly, Mamba [22], improves with a selective scanning mechanism and hardware-aware parallelism to enable very efficient training and inference, achieving comparable performances to Transformers on large-scale data. Yet, existing explorations focus on processing uni-modal data, and the multi-modal fusion mechanism on SSMs is still under-investigated.

In this paper, we observe that the explicit state variables in SSMs provide great fusion anchors, i.e., from which we can extract inter-model complementary information, and to where we can fuse the complementary information into a unified representation. Inspired by the effective Coupled Hidden Markov Model (CHMM) [23], we investigate the multi-modal fusion problem of the Mamba model from a state transition perspective. For multi-modal fusion on Mamba, the brute-force way is to direct aggregate features from all multi-modalities into one feature, i.e., the aggregation approach, and process with a sole Mamba model. However, such an approach neglects the individual intra-modal propagation. Instead, we propose the Coupled Mamba model, for coupling state chains of multiple modalities while maintaining independence of intra-modality state processes. Specifically, in our coupled scheme, we

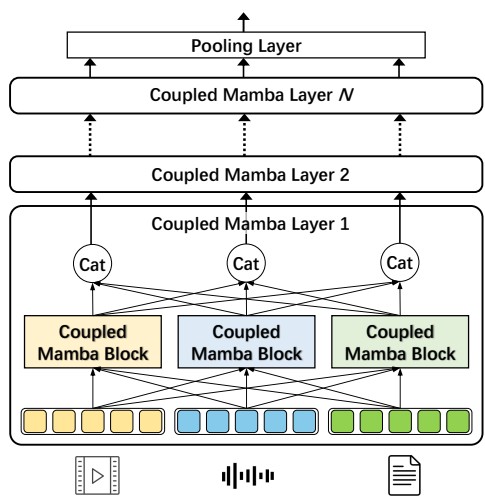

Figure 1: Architecture of Coupled Mamba.

devise an inter-modal hidden states transition scheme, in which the current state is dependent on the states of its own chain and that of the neighbouring chains at the previous time-step. Another challenge is to fully comply with the hardware-aware parallelism for efficiency, we achieve parallel computing by deriving multi-modal global convolution kernels. As shown in Figure 1, the entire Coupled Mamba model consists of $N$ layers, and is finally adapted to downstream tasks through pooling. Each layer has $M$ Coupled Mamba blocks, where $M$ is the number of modalities. Each Coupled Mamba block receives sequence data of multiple modalities as input, aggregate states from multiple modalities, and then transits into the state at next time of each individual modality. We conduct extensive experiments on CMU-MOSEI, CH-SIMS [24], CH-SIMSV2 [25] datasets through multi-domain input, and verify the effectiveness of our model compared to current state-of-the-art methods, with $0.4\%, 0.9\%, 2.3\%$ F1-Score increase, $49\%$ faster inference and $83.7\%$ GPU memory save. The results demonstrate that our Coupled Mamba model enhances the multi-modal fusion with state coupling.

## 2  Related Work

**Multi-modal Fusion** Multi-modal fusion focuses on combining features from various modalities into unified representations to tackle multi-modal learning challenges. Traditionally, fusion methods are categorized into feature-level early fusion and decision-level late fusion, based on where fusion occurs within the model [26]. Early fusion techniques are employed by [27] to merge features from diverse modalities such as audio, text, and vision. [28] introduce a method using two separate branches for spatial and temporal modalities with a straightforward post-fusion for video action recognition. Other notable post-fusion approaches include works like [9] and [29, 30], which suggest robust late fusion via rank minimization. Recent advances in deep learning have expanded the concept of early fusion to mid-term fusion, which integrates features at multiple levels [31]. For instance, [32] develop a fused representation by progressively combining multiple fusion layers. Similarly, [33] propose a multi-layer fusion method that connects all modality-specific networks through a central network. [**?** ] introduce an architecture search algorithm to identify the optimal fusion architecture. Furthermore,

[34, 35] incorporate attention mechanisms into multi-modal fusion, while [13] suggest exchanging feature channels between modalities. Additionally, [36] integrate bilinear pooling into attention blocks, showing its effectiveness in capturing higher-level feature interactions by stacking multiple attention blocks for image captioning. The focus has recently shifted towards dynamic fusion, which selects the optimal fusion strategy from various candidate operations based on inputs from different modalities [37, 38]. This dynamic approach offers greater flexibility for different multi-modal tasks compared to static methods. Inspired by the success of dynamic fusion designs and higher-level feature interaction capture in multi-modal fusion, our work aims to dynamically capture hidden states both within and between modalities using coupled state space models via state diffusion, enabling more efficient modality fusion for complex multi-modal tasks.

**State Space Models** State Space Models (SSM) are exceptionally effective at learning the complex correlations inherent in language sequences. The seminal work of [17] introduced the structured state space model (S4), which aims to encapsulate the extended dependency characteristics of language sequences. Conceptually, S4 combines the unique properties of CNNs and RNNs to create a powerful framework for sequential data processing. Building on the foundation laid by S4, subsequent research efforts have been devoted to solving the problem of linearly scaling sequence lengths. In this regard, [39] introduced S5 utilizing MIMO-SSM and parallel scan technology, while [40] proposed H3, which greatly improved the performance of SSM, and [18] introduced GSS, which demonstrated faster training and competitive performance. Furthering the current state of research, [22] developed a novel language model called Mamba. This model uniquely combines a data-selective SSM layer and a parallel scanning algorithm to solve Transformer's quadratic complexity calculation problem in long sequence modeling and Transformer's inability to model data outside the attention window. This also illustrates the huge potential of Mamba in processing sequence data.

**Coupled Hidden Markov Model** Hidden Markov Model (HMM) is a probabilistic model that simulates a sequence of hidden states to generate a sequence of observations. The core components of the model include the state transition matrix A, the observation probability matrix (emission matrix) B and the initial state probability vector $\pi$. This model assumes the existence of Markov chains between hidden states, and observation events are independently generated by hidden states. To address specific needs, researchers have developed several HMM variants. For example, the Hierarchical Hidden Markov Model (HHMM) [41] introduces a state hierarchy based on standard HMMs, while the Mixed Hidden Markov Model (MHMM) [42] combines multiple HMMs to Build complex distributions. These extensions improve the applicability of HMM in various scenarios and further promote the application of sequence data analysis in multiple fields. Coupled Hidden Markov Models (CHMM) [23] are a class of tools capable of modeling multiple interrelated time series. In multimodal fusion, we usually focus on signals from different channels, such as audio, text, and facial expressions, which are all time-correlated. Coupled HMMs can effectively model such data because they can consider dynamic correlations between multiple channels simultaneously.

## 3 Coupled State Space Model

In this section, we introduce Coupled Mamba method for multi-modal fusion in detail, which performs multi-modal fusion by introducing multi-modal historical states. As shown in Figure 2, it contains two parts: state coupling and state space model.

### 3.1 Preliminary

In recent years, the state space model has developed rapidly [17, 19, 40]. Mamba introduced a selectivity mechanism based on S4, which converted the original time-invariant characteristics. Mamba is based on the concept of continuous systems by introducing hidden states $h(t) \in \mathbb{R}^N$ to map a series of inputs $x(t) \in \mathbb{R}^L$ to obtain output $y(t) \in \mathbb{R}^L$, where N denotes the number of hidden states. The continuous system can be expressed as:

$$h'(t) = \mathbf{A}h(t) + \mathbf{B}x(t), \ \ y(t) = \mathbf{C}h(t). \tag{1}$$

where $\mathbf{A} \in \mathbb{R}^{N \times N}$ represents the state transition matrix of the system, and $\mathbf{B} \in \mathbb{R}^{N \times 1}, \mathbf{C} \in \mathbb{R}^{N \times 1}$ are projection matrices. Mamba uses a time scale parameter $\Delta$ to discretize the continuous parameters $\mathbf{A}, \mathbf{B}$ into $\overline{\mathbf{A}}, \overline{\mathbf{B}}$, the zero-order hold (ZOH) principle is adopted by default. The discretized state-space equation is:

$$\overline{\mathbf{A}} = \exp(\mathbf{\Delta A}), \ \ \overline{\mathbf{B}} = (\mathbf{\Delta A})^{-1}(\exp(\mathbf{\Delta A}) - \mathbf{I}) \cdot \mathbf{\Delta B}. \tag{2}$$

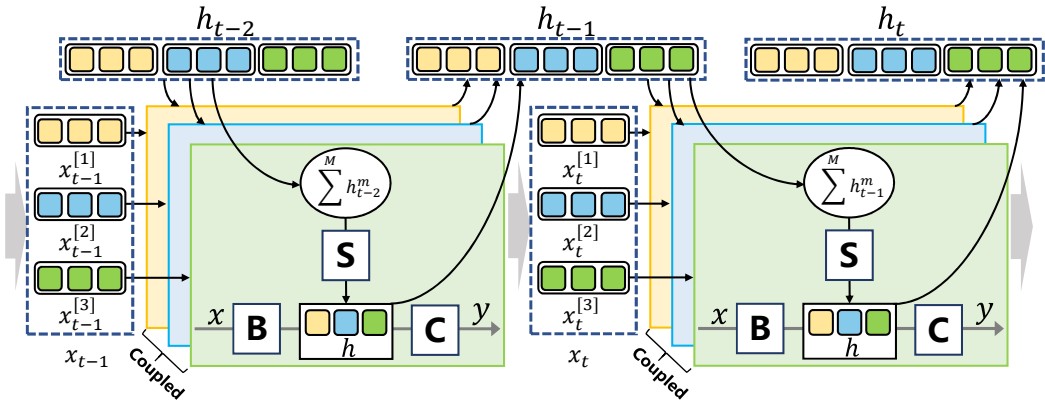

Figure 2: Coupling Mamba receives input $x_{t-1}$, and performs internal state switching and output through three key parameter matrices, where $\mathbf{B}, \mathbf{C}$ and $\mathbf{S}$ are respectively represented as the input matrix, output matrix and state transfer matrix. The hidden states are summed across modalities and used for state transition input to generate next time states. The state is propagated sequentially in time.

Then the discretized version of Eq. (1) with step size $\Delta$ can be rewritten as:

$$h_t = \overline{\mathbf{A}}h_{t-1} + \overline{\mathbf{B}}x_t, \ \ y_t = \mathbf{C}h_t. \tag{3}$$

Finally, by expanding $h_{t-1}$ layer by layer, the global convolution kernel $\overline{\mathbf{K}} \in \mathbb{R}^L$ can be obtained, and $\overline{\mathbf{K}}$ is used to calculate the output $y$, which is defined as follows:

$$\overline{\mathbf{K}} = \left( \mathbf{C}\overline{\mathbf{B}}, \mathbf{C}\overline{\mathbf{A}}\overline{\mathbf{B}}, ..., \mathbf{C}\overline{\mathbf{A}}^{\mathrm{L}-1}\overline{\mathbf{B}} \right),$$
$$y = x \otimes \overline{\mathbf{K}}. \tag{4}$$

where L is the length of the input sequence $x$ and $\otimes$ denotes the convolution operation. For algorithm 1, L denotes the sequence length, E denotes the extended dimension, D denotes the feature dimension, and B denotes the batch size.

### 3.2 Coupled State Transition

For multi-modality data input, one naive way is to aggregate the multi-modal features into one feature and process using a single Mamba model. However, such approach neglects intra-modal propagation. Inspired by the Coupled Hidden Markov Model (CHMM) [23], a more elegant solution is to model mutual modality transition probability as follows:

$$P_{i=1:M,j} = P\left( h_t^j | h_{t-1}^1, h_{t-1}^2, ..., h_{t-1}^M \right)$$

where $P_{i=1:M,j}$ is the probability transition matrix from all modalities to current modality $j$. For SSM with $M$ multi-modal input, we have $M$ state propagation sequences. In alignment with CHMM, we can model the state transition of a modality $m$ by coupling all the modality states as:

---

**Algorithm 1:** Coupled Mamba

**Data:** Input:
    $\mathbf{H_{t-1}} = \left\{ h_{t-1}^1, h_{t-1}^2, ..., h_{t-1}^M \right\}, \mathbf{x_{t-1}}:$
    $h_{t-1}^m \in \mathbb{R}^N, x_{t-1} \in (B, L, D)$
**Result:** Output: $\mathbf{y_t}: (B, L, D)$
**Require :** Input
**Ensure  :** Output

1  ;
2  Normalize the input sequence:
3  $\mathbf{x'_{t-1}}: (B, L, D) \leftarrow \mathbf{LayerNorm}(x_{t-1})$;
4  $\mathbf{u}: (B, L, E) \leftarrow \mathbf{Linear}_u(x'_{t-1})$;
5  $\mathbf{z}: (B, L, E) \leftarrow \mathbf{Linear}_z(x'_{t-1})$;
6  ;
7  Process with Coupled Mamba:
8  **for** $o$ in $forward$ **do**
9     $\mathbf{u'_o}: (B, L, E) \leftarrow \mathbf{SiLU}(\mathrm{Conv1d}_o(u))$;
10    $\mathbf{B_o}: (B, L, N) \leftarrow \mathbf{Linear}_B^o(u'_o)$;
11    $\mathbf{C_o}: (B, L, N) \leftarrow \mathbf{Linear}_C^o(u'_o)$;
12    $\mathbf{\Delta_o}: (B, L, E) \leftarrow \log(1 +$
      $\exp(\mathbf{Linear}_{\Delta_o}(u'_o) + \mathbf{Parameter}_{\Delta_o}))$;
13    $\mathbf{S_o}: (B, L, E, N) \leftarrow \mathbf{\Delta_o^N} \otimes \mathbf{Parameter}_{A_o}$;
14    $\mathbf{B_o}: (B, L, E, N) \leftarrow \mathbf{\Delta_o^N} \otimes \mathbf{B_o}$;
15    $\mathbf{y_o}: (B, L, E) \leftarrow$
      $\mathbf{CSSM}(\mathbf{S_o}, \mathbf{B_o}, \mathbf{C_o})(\mathbf{H_{t-1}}, \mathbf{u'_o})$;
16 **end**
17  ;
18 Get gated $y_o$:;
19 $\mathbf{y'_{forward}}: (B, L, E) \leftarrow y_{forward} \odot \mathbf{SiLU}(z)$;
20 Residual connection:;
21 $\mathbf{y_t}: (B, L, D) \leftarrow \mathbf{Linear}_T(y'_{forward}) + \mathbf{x_{t-1}}$;

---

$$h_t^m = \sum \left( \overline{\mathbf{A}}_{1,m}h_{t-1}^1, \overline{\mathbf{A}}_{2,m}h_{t-1}^2, ..., \overline{\mathbf{A}}_{M,m}h_{t-1}^M \right) + \overline{\mathbf{B}}_m x_t^m, \ \ y_t^m = \mathbf{C}h_t^m. \tag{5}$$

where $\overline{\mathbf{A}}_{i,m}$ denotes the state transition matrix from modality $i$ to $m$.

Taking into account the memory overhead and computational efficiency, such modeling increase the number of parameters and computational complexity greatly. We propose a more memory efficient way by conducting summation before state transition, which achieves similar performance and is much more efficient. So our formation of Coupled SSM is:

$$h_t^m = \mathbf{S}_m \sum_{m=1}^{M} h_{t-1}^m + \overline{\mathbf{B}}_m x_t^m \tag{6}$$

Where we use $\mathbf{S}_m \in \mathbb{R}^{B \times L \times D \times N}$ to model the overall state transition after states summation. One minor drawback of this modeling is that we require all modalities to have the same state, which can be easily addressed by using projection layers.

### 3.3 Parallelism and Efficiency Analysis

The main difference between Mamba and traditional recurrent neural networks (RNNs) is that the transition between states does not rely on any activation function. This feature enables it to pre-calculate intermediate results through the iterative Eq.(3), thereby achieving parallel computing. However, Coupled Mamba adds multi-modal state information based on Mamba, which brings new challenges to the ability to maintain the Mamba parallelization algorithm. In order to solve this problem, we derived a global convolution kernel suitable for Coupled Mamba to ensure that Coupled Mamba can continue to enjoy the advantages brought by Mamba parallel computing, thereby effectively improving the throughput and inference speed of the model. Detailed analysis on throughput and inference speed will be discussed in depth in subsequent sections.

After introducing the state information of different modals, we learned about the entire state transfer process (6) through 3.2. By deriving Eq.(6), that is, disassembling $h_{t-1}^m$, we can get the following results:

$$\mathbf{P} = \sum_{m=1}^{M} \mathbf{S}_m, \quad \mathbf{U_t} = \sum_{m=1}^{M} \overline{\mathbf{B}}_m x_t^m, \quad h_t^m = \mathbf{S}_m \sum_{i=0}^{t-1} \mathbf{P}^i \mathbf{U}_{t-1-i} + \overline{\mathbf{B}}_m x_t^m. \tag{7}$$

where $\mathbf{P} \in \mathbb{R}^{B \times L \times D \times N}$. According to Eq.(7) which can be extended to the state information of each modal, we use the following formula to calculate the output.

$$y = \mathbf{C} \otimes \sum_{m=1}^{M} h_t^m = \mathbf{C} \otimes \sum_{i=0}^{t} \mathbf{U_i} \mathbf{P^{t-i}} \tag{8}$$

From this, the global convolution kernel $\overline{\mathbf{K}} = \left( \mathbf{CP^0}, \mathbf{CP^1}, ..., \mathbf{CP^{t-1}}, \mathbf{CP^t} \right)$ suitable for Coupled Mamba can be obtained.

The global convolution kernel $\overline{\mathbf{K}}$ can be used to perform convolution operations on sequence data. In the convolution operation, the calculations of each convolution kernel and the input sub-region are independent of each other, allowing parallel processing of different convolution kernels or input blocks.

## 4 Experiment

To evaluate the effectiveness of our proposed Coupled Mamba in multi-modal fusion, we conduct extensive experiments, with special focus on the multi-modal sentiment analysis (MSA) task as it relies heavily on multi-modal data and is in sequential form. The MSA task aims to predict people's emotional polarity by fusing audio, text, and visual information. To fully evaluate the advantages of our approach, we conduct extensive experiments on both classification and regression tasks.

### 4.1 Datasets and Implementation Details

**Datasets** We conduct experiments on five benchmark datasets (CMU-MOSEI, CH-SIMS [24], CH-SIMSV2 [25], MM-IMDB and BRCA). CMU-MOSEI dataset is an extension of CMU-MOSI,

contains 22856 samples of movie review video clips. In this dataset, 16326 samples are used as the training set, and the remaining 1871 and 4659 samples are used as the validation set and test set respectively. CH-SIMS contains 2281 video clip samples, 1368 samples are used as the training set, and the remaining 456 and 457 samples are used as the validation set and test set respectively. CH-SIMSV2 is an extension of CH-SIMS, which contains 4402 video clip samples, of which 2722 samples are used as the training set, and the remaining 647 and 1034 samples are used as the validation set and test set respectively. For the feature extraction method of the dataset, please refer to the Appendix for more information. The MM-IMDB dataset is used for the movie genre classification task, which classifies movies based on posters and text descriptions. The BRCA dataset includes mRNA expression, DNA methylation, and miRNA expression data for predicting PAM50 subtype classification of breast cancer.

**Evaluation metrics** For regression tasks, we use the mean absolute error (MAE), which is the average absolute difference between the predicted value and the true value, and the Pearson correlation coefficient (Corr), which measures the degree of deviation of the prediction according to the following formula: The positive/negative and non-negative/negative classification results calculate the binary classification accuracy (Acc-2) and F1-Score, where Acc-2 and F1-Score are more important indicators. For classification tasks, we use Acc-2, Acc-3 and F1-Score (Weighted-F1, Macro-F1, Micro-F1, F1-score3) as evaluation indicators. F1-score3 is the overall performance evaluation of all categories, and F1-score is the performance evaluation of two categories. At the same time, the neutral category is ignored. All experiments were conducted in the same environment.

**Implementation details** We use a hidden dimension size of 128, an expansion coefficient of 2, a convolution kernel size of 4, $\Delta = dstate/8$ as the configuration of each Mamba block, and a layer number of 3 to train our Coupled Mamba. We use Adam to optimize the model and set the learning rate to $0.0005$ , weight decay coefficient is 0.0005, epoch is 150, the batch size is set to 1024, 128, 256 on CMU-MOSEI, CH-SIMS, and CH-SIMSV2. L1 loss is used as the loss function for the regression task, and cross entropy is used as the loss function for the classification task. All experiments were conducted on a Linux workstation equipped with a single NVIDIA 32GB V100GPU and a 32-core Intel Xeon CPU. More experimental details can be found in the Appendix.

## 4.2 Comparison with the state-of-the-arts

To fully validate the performance of Coupled Mamba, we conduct extensive comparisons with the following baselines [43, 30, 27, 11, 44, 45] in Table 1. We ran five times and reported the average value. We use bold text to show the best results. Traditionally, models that use aligned corpora tend to perform better [27]. In our experiments, we achieve significant improvements on all evaluation metrics compared to unaligned models. Our unaligned method is able to achieve better results even when compared with aligned models.

Table 1: Results on CMU-MOSEI. All models are based on language features extracted by BERT. The one with ∗ indicates that the model reproduces under the same conditions.

| Model | CMU-MOSEI | | | | Data Setting |
|---|---|---|---|---|---|
| | $MAE \downarrow$ | $Corr \uparrow$ | $Acc-2 \uparrow$ | $F1-Score \uparrow$ | |
| TFN [9] | 0.593 | 0.700 | 82.5 | 82.1 | Unaligned |
| LMF [30] | 0.623 | 0.677 | 82.0 | 82.1 | Unaligned |
| MFN [10] | - | - | 76.0 | 76.0 | Aligned |
| MFM [46] | 0.568 | 0.717 | 84.4 | 84.3 | Aligned |
| MulT [27] | 0.580 | 0.703 | 82.5 | 82.3 | Aligned |
| MAG-BERT [47] | - | - | 84.7 | 84.5 | Aligned |
| ICCN [48] | 0.565 | 0.713 | 84.2 | 84.2 | Aligned |
| MISA [11] | 0.555 | 0.756 | 85.5 | 85.3 | Aligned |
| TETFN [45] | 0.551 | 0.748 | 85.1 | 85.2 | Unaligned |
| DMD [44] | - | - | 84.8 | 84.7 | Unaligned |
| IMDer3 [43] | - | - | 85.1 | 85.1 | Unaligned |
| MAG-BERT∗ [47] | 0.549 | 0.753 | 85.2 | 85.1 | Aligned |
| **Coupled Mamba (Ours)** | **0.547** | **0.756** | **85.6** | **85.5** | Unaligned |
| **Coupled Mamba (Ours)** | **0.547** | **0.758** | **85.7** | **85.6** | Aligned |

In multi-modal sentiment analysis tasks, language is a key factor because different languages may have different ways of expressing the same emotion. However, Table 2 shows that our Coupled Mamba shows robustness in both English and Chinese sentiment analysis tasks. Even with unaligned data, our method still achieves highest performance.

The results of the classification task are given in Table 3 4. It can be seen from the results of the 1 that our proposed fusion method achieves state-of-the-art (SOTA) regardless of whether the data are aligned and from the results of the 4, we find that Coupled Mamba also performs well on Chinese datasets. This is sufficient to demonstrate the effectiveness and robustness of our method.

Table 2: Results on CH-SIMS (Chinese). All models are based on language features extracted by BERT, and the results are compared on unaligned data. Acc-N represents N-level accuracy.

| Model | CH-SIMS | | | | |
|---|---|---|---|---|---|
| | $Acc-2\uparrow$ | $Acc-3\uparrow$ | $Acc-5\uparrow$ | $F1-Score\uparrow$ | $MAE\downarrow$ |
| TFN [9] | 78.4 | 65.1 | 39.3 | 78.6 | 0.432 |
| LMF [30] | 77.8 | 64.7 | 40.5 | 77.9 | 0.411 |
| MFN [10] | 77.9 | 65.7 | 39.5 | 77.9 | 0.435 |
| MulT [27] | 78.6 | 64.8 | 37.9 | 79.7 | 0.453 |
| Self-MM [8] | 80.0 | 65.5 | 41.5 | 80.4 | 0.425 |
| TETFN [45] | 81.2 | 63.2 | 41.8 | 80.2 | 0.420 |
| IMDer [43] | 76.3 | - | **50.7** | 76.4 | - |
| **Coupled Mamba (Ours)** | **81.8** | **68.7** | 42.1 | **81.3** | **0.409** |

Table 3: Results of classification tasks on CMU-MOSEI. All models are based on language features extracted by BERT, and the results are performed on unaligned data. We ran it five times and report the average results.

| Model | CMU-MOSEI | | | |
|---|---|---|---|---|
| | $Acc-2\uparrow$ | $Acc-3\uparrow$ | $F1-Score\uparrow$ | $F1-Score-3\uparrow$ |
| EF-LSTM [49] | 26.73 | 66.09 | 28.12 | 63.68 |
| Graph-MFN [50] | 28.47 | 66.39 | 28.77 | 64.00 |
| TFN [9] | 28.66 | 66.63 | 28.75 | 63.93 |
| LMF [30] | 28.66 | 66.59 | 28.92 | 64.86 |
| MFN [10] | 28.61 | 66.59 | 28.70 | 64.31 |
| MulT [27] | 27.38 | 67.04 | 28.67 | 65.01 |
| MISA [11] | 28.50 | 67.63 | 29.03 | 65.39 |
| Self-MM [8] | 29.67 | 68.15 | 28.86 | 66.53 |
| TETFN [45] | 29.54 | 67.95 | 28.47 | 66.33 |
| **Coupled Mamba (Ours)** | **32.02** | **68.95** | **29.72** | **67.76** |

Table 4: Classification task results on CH-SIMS. All models are based on language features extracted by BERT and the results are performed on unaligned data. We ran it five times and report the average results.

| Model | CH-SIMS | | | |
|---|---|---|---|---|
| | $Acc-2\uparrow$ | $Acc-3\uparrow$ | $F1-Score\uparrow$ | $F1-Score-3\uparrow$ |
| EF-LSTM [49] | 56.27 | 54.27 | 49.85 | 38.18 |
| Graph-MFN [50] | 57.99 | 68.44 | 54.66 | 63.44 |
| TFN [9] | 53.56 | 65.95 | 52.79 | 62.04 |
| LMF [30] | 57.06 | 66.87 | 53.83 | 62.46 |
| MFN [10] | 56.96 | 67.57 | 54.14 | 62.37 |
| MulT [27] | 56.34 | 68.27 | 54.26 | 64.23 |
| MISA [11] | 57.27 | 67.05 | 53.99 | 60.98 |
| Self-MM [8] | 58.65 | 67.56 | 55.88 | 65.95 |
| TETFN [45] | 57.77 | 66.83 | 55.15 | 65.23 |
| **Coupled Mamba(Ours)** | **60.12** | **68.75** | **56.15** | **67.47** |

Table 7 shows the results on the CH-SIMSV2 dataset, which currently only supports regression tasks. It can be seen from the table that the method we proposed has achieved a huge improvement of $2.3\%, 2,3\%$ in Acc-2 and F1-Score respectively, indicating the effectiveness of our method.

The results of Coupled Mamba on BRCA and MM-IMDB datasets are shown in Table 5 6. Whether in multimodal sentiment analysis tasks or in movie genre classification tasks or biology classification

Table 5: Result on the BRCA benchmark: mR, D, and miR denote mRNA expression, DNA methylation, and miRNA expression data respectively. The best results are in bold.

| | Modality | Acc(%)↑ | WeightedF1(%)↑ | MacroF1(%)↑ |
|---|---|---|---|---|
| GRridg [51] | mR+D+miR | 74.5 | 72.6 | 65.6 |
| GMU [52] | mR+D+miR | 80.0 | 79.8 | 74.6 |
| CF [53] | mR+D+miR | 81.5 | 81.5 | 77.1 |
| MOGONET [54] | mR+D+miR | 82.9 | 82.5 | 77.4 |
| TMC [55] | mR+D+miR | 84.2 | 84.4 | 80.6 |
| MM-Dynamics [56] | mR+D+miR | 87.5 | 87.6 | 83.9 |
| **Coupled Mamba(Ours)** | mR+D+miR | **88.1** | **88.5** | **85.4** |

Table 6: Result on the MM-IMDB benchmark. **I** and **T** denote image and text respectively. The best results are in bold.

| | Modality | MicroF1(%)↑ | MacroF1(%)↑ |
|---|---|---|---|
| LRMF [57] | I+T | 58.95 | 50.73 |
| MFM [46] | I+T | 56.44 | 48.53 |
| MI-Matrix [58] | I+T | 55.87 | 46.77 |
| RMFE [59] | I+T | 58.67 | 49.82 |
| CCA [60] | I+T | 60.31 | 50.45 |
| RefNet [61] | I+T | 59.45 | 51.51 |
| DynMM [62] | I+T | 60.35 | 51.60 |
| **Coupled Mamba (Ours)** | I+T | **62.41** | **52.58** |

tasks, Coupled Mamba can show excellent performance. We expect that coupled mamba can be extended to other multimodal tasks.

Table 7: Results on CH-SIMSV2, consistent across all experimental settings, using unaligned data. We run it five times and report the average results.

| Model | CH-SIMSV2 | | | | | |
|---|---|---|---|---|---|---|
| | $Acc-2\uparrow$ | $Acc-3\uparrow$ | $Acc-5\uparrow$ | $F1-Score\uparrow$ | $MAE\downarrow$ | $Corr\uparrow$ |
| TFN [9] | 80.1 | 72.3 | 52.5 | 80.1 | 30.3 | 70.7 |
| LMF [30] | 74.1 | 64.9 | 47.8 | 73.8 | 36.7 | 55.7 |
| MFN [10] | 81.1 | 73.7 | 54.5 | 81.2 | 29.5 | 72.6 |
| MulT [27] | 80.7 | 73.1 | 54.8 | 80.7 | 29.1 | 73.8 |
| MAG-BERT [47] | 79.8 | 73.5 | 53.7 | 79.8 | 33.4 | 69.1 |
| Self-MM [8] | 79.7 | 72.6 | 52.8 | 79.7 | 31.1 | 69.5 |
| TETFN [45] | 79.7 | 73.6 | 54.4 | 79.8 | 31.1 | 69.5 |
| **Coupled Mamba** | **83.4** | **75.0** | **55.1** | **83.5** | **28.7** | **75.8** |

### 4.3 Ablation study

We evaluated the impact of each component in Coupled Mamba to verify the effectiveness of our design. It is worth noting that in order to reduce the impact of randomness on the experimental results, our entire ablation experiment was conducted on the CMU-MOSEI dataset.

We use the cross-attention mechanism instead of the fusion strategy for comparison. The results are shown in Table 8. Coupled Mamba filters input through a selective mechanism and uses historical modal information to remember and perceive global context, so Coupled Mamba also performs modal fusion well on unaligned data. In contrast, cross-attention is sensitive to misaligned data, and this spatio-temporal inconsistency will lead to insufficient integration between modalities and poor performance.

Table 8: All things being equal, replacing Coupled Mamba with Cross attention, we execute it five times and report the average results.

| Method | CMU-MOSEI | | | | Data Setting |
| --- | --- | --- | --- | --- | --- |
| | $MAE \downarrow$ | $Corr \uparrow$ | $Acc - 2 \uparrow$ | $F1 - Score \uparrow$ | |
| Cross Attention | 55.9 | 73.3 | 84.6 | 84.5 | Unaligned |
| **Coupled Mamba (Ours)** | **54.7** | **75.6** | **85.6** | **85.5** | Unaligned |

The number of hidden states and size of $\Delta$ will have an impact on the results. The size of $\Delta$ affects SSM's ability to retain historical information. An increased size of $\Delta$ focuses more on the present input while disregarding past data, and it also raises the count of hidden states. This escalation in complexity might result in overfitting, higher computational expenses, and may not enhance the model's actual effectiveness. In order to explore the impact of these parameters on the results, we conducted multiple experiments, and the experimental results showed that the model performance changed under different parameter settings. Detailed results can be found in Tables 9, 10. With $\Delta = dstate/8$ and $dstate = 64$, Coupled Mamba achieves the best performance than other configurations.

Table 9: Performance on CMU-MOSEI with different timescale $\Delta$

| $\Delta$ | CMU-MOSEI | | |
| --- | --- | --- | --- |
| | Corr↑ | Acc-2↑ | F1-Score↑ |
| $dstate/16$ | 75.3 | 85.2 | 85.0 |
| $dstate/8$ | **75.6** | **85.6** | **85.5** |
| $dstate/4$ | 74.2 | 85.0 | 84.9 |

Table 10: Performance on CMU-MOSEI with different $dstate$

| $dstate$ | CMU-MOSEI | | |
| --- | --- | --- | --- |
| | Corr↑ | Acc-2↑ | F1-Score↑ |
| 128 | 74.1 | 84.2 | 84.1 |
| 64 | **75.6** | **85.6** | **85.5** |
| 32 | 75.0 | 84.9 | 84.9 |

In order to verify the effectiveness of our state coupling, we adopt the splicing fusion, average fusion, and native Mamba blocks for experiments. Average Fusion and Concat Fusion refer to averaging and concatenating the features of different modalities and then sending them to a single Mamba Block for processing. Mamba Fusion refers to using a Mamba Block to process each modality, and finally weighting the results of the three blocks for downstream tasks. The result is shown in Table 11, our Coupled Fusion obtains the best performance than others. Traditional modal fusion methods, such as averaging and concatenation, fail to fully cope with the inherent heterogeneity of multi-modal data. Such methods ignore the different influences that different modalities may have on specific tasks, thereby failing to effectively reveal the intrinsic correlation between multi-modal data. Simple Mamba blocks are not enough to dynamically grasp the semantic relationships. The introduction of state coupling mechanism based on Mamba can make up for this shortcoming and achieve significant improvements in multiple performance indicators.

Table 11: Comparison of fusion methods

| Model | CMU-MOSEI | | | |
| --- | --- | --- | --- | --- |
| | $MAE \downarrow$ | $Corr \uparrow$ | $Acc - 2 \uparrow$ | $F1 - Score \uparrow$ |
| Average Fusion | 56.4 | 73.6 | 84.2 | 84.1 |
| Concat Fusion | 56.2 | 72.8 | 84.8 | 84.5 |
| Mamba Fusion | 55.3 | 74.9 | 85.3 | 85.3 |
| **Coupled Fusion** | **54.7** | **75.6** | **85.6** | **85.5** |

Compared to Transformers, our approach improves performance by $1\% \sim 2\%$ as shown in Table 8 and decreases memory consumption by more than **83. 7%** for sequences length 500 according to Figure 3. When the sequence length increases, the GPU memory usage of the Transformer-based method increases exponentially. In comparison, our method exhibits linear growth. As the sequence grows, the advantages of Coupled Mambas become more apparent.

As shown in Figure 4, we compared Coupled Mamba and Transformers with five different sequence lengths, and the results show that our inference speed is twice as fast as Transformers under the same sequence length. However, as the sequence length continues to grow, the inference speed of Coupled Mamba will far exceed that of Transformers.

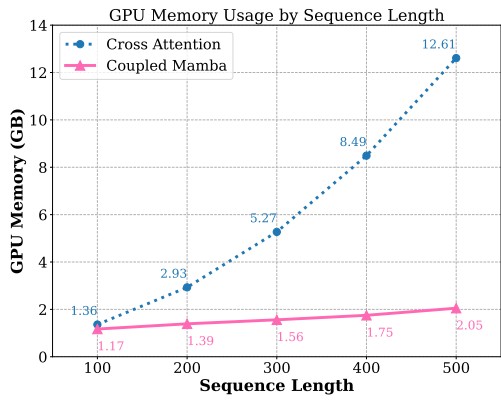

Figure 3: GPU usage comparison

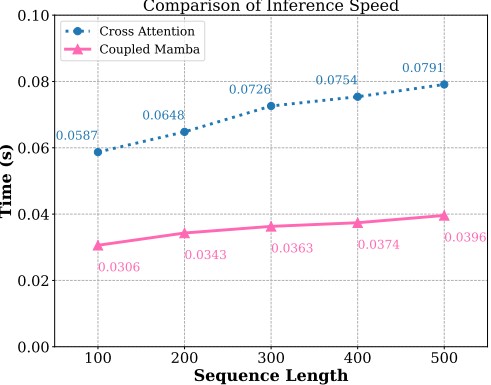

Figure 4: Inference speed comparison

To verify the robustness of our proposed method, we conducted experiments on the CMU-MOSEI dataset with missing data. Specifically, we created a random mask with the same shape as the original tensor, where each element is taken from the Bernoulli distribution B(1-p). This means that each element has a p% probability of being 1 (retained) and a (1-p)% probability (i.e., the missing rate (MR)) of being 0 (missing). We then multiplied this random mask with the original tensor so that the regions with masked values of 0 result in missing data in the original tensor.

Table 12: Performance of Coupled Mamba on CMU-MOSEI dataset when data is missing. Other baselines are from [63]

| MR | DCCA [64] | DCCAE [65] | MCTN [66] | MMIN [67] | GCNET [68] | Coupled Mamba |
|---|---|---|---|---|---|---|
| 0.0 | 80.7/80.9 | 81.2/81.2 | 84.2/84.2 | 84.3/84.2 | 85.2/85.1 | **85.5/85.6** |
| 0.1 | 77.4/77.3 | 78.4/78.3 | 81.8/81.6 | 81.9/81.3 | 82.3/82.1 | **82.6/82.7** |
| 0.2 | 73.8/74.0 | 75.5/75.4 | 79.0/78.7 | 79.8/78.8 | 80.3/79.9 | **81.1/80.9** |
| 0.3 | 71.1/71.2 | 72.3/72.2 | 76.9/76.2 | 77.2/75.5 | 77.5/76.8 | **81.0/81.0** |
| 0.4 | 69.5/69.4 | 70.3/70.0 | 74.3/74.1 | 75.2/72.6 | 76.0/74.9 | **78.4/78.5** |
| 0.5 | 67.5/65.4 | 69.2/66.4 | 73.6/72.6 | 73.9/70.7 | 74.9/73.2 | **77.4/77.7** |
| 0.6 | 66.2/63.1 | 67.6/63.2 | 73.2/71.1 | 73.2/70.3 | 74.1/72.1 | **75.1/75.4** |
| 0.7 | 65.6/61.0 | 66.6/62.6 | 72.7/70.5 | 73.1/69.5 | 73.2/70.4 | **74.1/74.2** |
| Average | 70.3/71.2 | 72.6/71.2 | 77.0/76.1 | 77.3/75.4 | 77.9/76.8 | **79.4/79.5** |

The results are shown in Table 12, and the numbers of other baselines are from [63]. Our method shows the best performance. Note that the left side of / shows Acc-2, while the right side indicates the F1-score.

## 5 Conclusion and Discussion

In this paper, we introduce Coupled Mamba, a novel approach to enhance multi-modal fusion by leveraging state evolution chains within state space. Our method integrates intermediate information from various modalities, capturing dynamic multi-modal interactions over time. This addresses challenges in parallel SSM with multiple inputs. Both quantitative and qualitative experiments confirm the effectiveness of Coupled Mamba. Code is available at https://github.com/hustcselwb/coupled-mamba.

## Acknowledgments

This work is supported by the National Natural Science Foundation of China (NSFC No. 62272184 and No. 62402189), the China Postdoctoral Science Foundation under Grant Number GZC20230894, and the China Postdoctoral Science Foundation (Certificate Number: 2024M751012). The computation is completed in the HPC Platform of Huazhong University of Science and Technology.

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

# A  Appendix / supplemental material

The following content is the entire process of pushing to the Coupled Mamba parallelization guarantee. First, we first define the symbols. Assume that the total number of modes is M, $h_{t-1}^m$ represents the hidden state at time t-1, where m is any mode, $\mathbf{A} \in \mathbf{R}^{D \times N}$ represents the state transition matrix, $\mathbf{B} \in \mathbf{R}^{B \times L \times N}$ represents the selective matrix obtained by mapping from the current input, $\mathbf{C} \in \mathbf{R}^{B \times L \times N}$ is the same as $\mathbf{B}$, where superscript $B$ is the batch size, $L$ is the input time series length, and $N$ is the number of hidden states.

Since the equations of the state space model and Mamba's core processes, such as discretization processing, hardware-aware algorithms, and parallel execution theory, have been discussed in the text, they will not be repeated below.

The core of Coupled Mamba is to introduce multi-modal information while ensuring the parallel computing advantages of Mamba block. After introducing multi-modal information, we have:

$$h_t^m = \overline{A}_m G_m \sum_{m=1}^{M} h_{t-1}^m + \overline{B}_m X_t^m \tag{9}$$

where $\overline{A}_m \in \mathbb{R}^{B \times L \times D \times N}$, $G_m \in \mathbb{R}^{N \times N}$ is a coupling matrix, which can be understood as a shared state transition matrix, which transfers the coupling state based on a certain probability based on the comprehensive state at time $t-1$. By integrating $\mathbf{G_m}$ into $\overline{\mathbf{A}}_{\mathbf{m}}$, we can get its unified representation $\mathbf{S_m} \in \mathbb{R}^{B \times L \times D \times N}$. Therefore, the above formula can be expressed as

$$h_t^m = \mathbf{S_m} \sum_{m=1}^{M} h_{t-1}^m + \overline{B}_m X_t^m \tag{10}$$

Next, let us derive it step by step starting from time 0, when $t = 0$ we have:

$$h_0^m = \overline{B}_m x_0^m \tag{11}$$

when $t = 1$, we can get:

$$h_{t=1}^1 = S_1 \sum_{m=1}^{M} h_{t=0}^m + \overline{B}_1 x_{t=1}^1 \tag{12}$$

$$h_{t=1}^2 = S_2 \sum_{m=1}^{M} h_{t=0}^m + \overline{B}_2 x_{t=1}^2 \tag{13}$$

The subscripts of $\mathbf{S}$ and $\overline{\mathbf{B}}$ represent different modalities, and the superscript of $x_t^m$ represents different modalities. Through this recursive formula we can get

$$h_{t=1}^M = S_M \sum_{m=1}^{M} h_{t=0}^m + \overline{B}_M x_{t=1}^M \tag{14}$$

In the same way, we bring $h_{t=1}^1, h_{t=1}^2, ..., h_{t=1}^M$ into the formula for calculating each mode $h_{t=2}^m$, we can get:

$$h_{t=2}^1 = S_1 \sum_{m=1}^{M} h_{t=1}^m + \overline{B}_1 x_{t=2}^1 \tag{15}$$

By disassembling $h_{t=1}^1, h_{t=1}^2, ..., h_{t=1}^M$, we can get:

$$h_{t=2}^1 = S_1 \left( \sum_{m=1}^{M} h_0^m \sum_{m=1}^{M} S_m + \overline{B}_1 x_{t=1}^1 + \overline{B}_2 x_{t=1}^2 + .... + \overline{B}_M x_{t=1}^M \right) \tag{16}$$

where

$$\sum_{m=1}^{M} h_0^m = \overline{B}_1 x_{t=0}^1 + \overline{B}_2 x_{t=0}^2 + ... + \overline{B}_M x_{t=0}^M \tag{17}$$

Let $\overline{B}_1 x_t^1 + \overline{B}_2 x_t^1 + ... + \overline{B}_M x_t^1$ be $U_t$, $P = \sum_{m=1}^{M} S_m$ ,$U_t$ can be expressed as

$$U_t = \sum_{m=1}^{M} \overline{B}_m x_t^m \tag{18}$$

Similarly we can get

$$h_{t=2}^1 = S_1 \left( PU_0 + U_1 \right) + \overline{B}_1 x_{t=2}^1 \tag{19}$$

$$h_{t=3}^1 = S_1 \left( P^2 U_0 + PU_1 + U_2 \right) + \overline{B}_1 x_{t=3}^1 \tag{20}$$

Therefore, by recursively recursing this formula, we can get

$$h_t^m = S_m \sum_{i=0}^{t-1} P^i U_{t-1-i} + \overline{B}_m x_t^m \tag{21}$$

Finally we calculate the output through $y = \mathbf{C} \otimes \sum_{m}^{M} h_t^m$, we have

$$y = \mathbf{C} \otimes \sum_{i=0}^{t} \mathbf{U_i} \mathbf{P^{t-i}} \tag{22}$$

where $\otimes$ represents the convolution operation, and the convolution kernel is $\overline{\mathbf{K}} = \left( \mathbf{CP^0}, \mathbf{CP^1}, ..., \mathbf{CP^{t-1}}, \mathbf{CP^t} \right)$. At this point, we have completed the derivation of the entire parallelized calculation.

In order to fully verify the effectiveness of this research method, we further performed classification experiments on the CMU-MOSI. The experimental results are displayed in Table 13, as follows:

Table 13: Results on the CMU-MOSI dataset for classification task, all results are performed under the same conditions, and the average results are reported after five runs.

| Model | CMU-MOSI | | | |
| --- | --- | --- | --- | --- |
| | Acc-2↑ | Acc-3↑ | F1-Score↑ | F1-Score-3↑ |
| EF-LSTM [49] | 46.40 | 74.43 | 46.55 | 72.74 |
| Graph-MFN [50] | 46.13 | 75.34 | 46.96 | 73.71 |
| TFN [9] | 43.78 | 73.44 | 45.55 | 71.86 |
| LMF [30] | 45.76 | 74.11 | 46.34 | 72.50 |
| MFN [10] | 46.68 | 74.99 | 46.78 | 74.22 |
| MulT [27] | 48.93 | 74.99 | 47.29 | 73.06 |
| MISA [11] | 44.15 | 76.30 | 46.75 | 74.57 |
| Self-MM[8] | 51.48 | 77.74 | 48.37 | 76.25 |
| TETFN [45] | 50.74 | 77.67 | 47.88 | 75.85 |
| **Coupled Mamba (Ours)** | **53.76** | **78.59** | **49.21** | **76.76** |

The CH-SIMSV2 [25] dataset is a Chinese data set for multi-modal sentiment analysis and is an extension of the CH-SIMS data set. The dataset contains audio, text, and video clips from different emotion categories, and each clip is labeled with emotional polarity, such as happy, sad, angry, etc. Each emotion category has corresponding speech, text, and video clips, as well as emotion labels associated with them.

**Feature extraction** CMU-MOSEI uses the pre-trained BERT model to extract language features and obtains 768-dimensional hidden states as word embeddings. For the visual modality, each video frame is encoded using Facet to represent the presence of a total of 35 facial action units. The acoustic model is processed by COVAREP to obtain 74-dimensional features. CH-SIMS uses pre-trained Chinese BERTbase word embeddings to obtain word vectors from text records, and finally represents each word as a 768-dimensional word vector. Acoustic features at 22050Hz were extracted using the LibROSA speech toolkit with default parameters. A total of 33-dimensional frame-level acoustic features are extracted. Extract aligned faces using MTCNN face detection algorithm. The MultiComp OpenFace2.0 toolkit was then used to extract a collection of 68 facial landmarks, 17 facial action

units, head pose, head orientation, and eye gaze. Finally, a total of 709-dimensional frame-level visual features were extracted.

**Exploring Coupled Mamba Layers** We investigated the number of layers in Coupled Mamba by performing experiments shown in Table 14. The optimal performance of our Coupled Mamba was observed at $layer = 3$.

Table 14: Performance on CMU-MOSEI with different $layers$

| Model | CMU-MOSEI | | | |
|---|---|---|---|---|
| | MAE↓ | Corr↑ | Acc-2↑ | F1-Score↑ |
| Coupled Mamba($Layer = 1$) | 55.1 | 74.7 | 84.7 | 84.8 |
| Coupled Mamba($Layer = 2$) | 55.3 | 74.9 | 84.9 | 84.8 |
| Coupled Mamba($Layer = 3$) | **54.7** | **75.6** | **85.6** | **85.5** |
| Coupled Mamba($Layer = 4$) | 56.6 | 74.5 | 84.8 | 84.7 |

