# OpenReview forum: "Coupled Mamba: Enhanced Multimodal Fusion with Coupled State Space Model"
_NeurIPS.cc/2024/Conference — NeurIPS 2024 poster_

### Official Review · Reviewer_nXDG · 2024-06-28

**Soundness:** 2
**Presentation:** 3
**Contribution:** 2
**Rating:** 5
**Confidence:** 5

**Summary:**

This paper extends the state space model Mamba into the multi-modal domain. The authors propose utilizing separate Mamba blocks to process each modality and suggest conditioning the state of each modality on the others to facilitate modality interaction. They further introduce a parallelism technique for the coupled Mamba transition. The framework is evaluated on three benchmark datasets.

**Strengths:**

1. The attempt to extending Mamba model to multi-modal domain is a motivating direction, which may facilitate the efficiency of future multimodal foundation model.

2. The model achieved good evaluation results on 3 benchmark datasets, while reducing the GPU memory usage to linear cost.

3. Detailed mathematical derivation of the new global convolutional kernel is provided.

**Weaknesses:**

1. The demonstration of the parallelism is not clearly illustrated. The original Mamba model's essence lies in the selection mechanism and the hardware-aware parallel scan. The parallelism achieved through the global convolution kernel in S4 is no longer as effective in the Mamba framework, given that the transition matrices at each time step are conditioned on the input. This inspired the introduction of the parallel scan method in the Mamba paper. While the authors show that coupled Mamba can be computed using a global convolution, the question remains, how is the parallelism achieved in the context of Mamba's scanning process?

2. The idea of using a sequential model to model each modality and then employing a simple interaction scheme (in this paper, sum) to enable modality fusion is straightforward, but maybe too simple to model the complex modality interaction.

3. Comparison with existing multi-modal Mamba models, for instance, [1-2], should be presented.

[1] VL-Mamba: Exploring State Space Models for Multimodal Learning
[2] Cobra: Extending Mamba to Multi-Modal Large Language Model for Efficient Inference

**Questions:**

1. In Table 9, does 'average (concat) fusion' means averaging (concating) the features from all modality and then go through a single mamba block?

**Limitations:**

Yes

---

> ### Author Rebuttal · Authors · 2024-08-07
>
> Thank you for your detailed and insightful assessment of our paper; we are deeply grateful for the time and expertise you dedicated.
>
> **W1: Parallelism through global convolution:**
>
> S4 relies on a global convolution kernel to achieve parallelism, while Mamba iterates on Equation 3 to compute intermediate results, ultimately accelerating computation using a global convolution kernel. In our Coupled Mamba, we retain the parallel scanning scheme within each modality. During the parallel scanning process, we iteratively combine intermediate results from each modality, integrating historical states across modalities. Simultaneously, we derive a multimodal global convolution kernel to fuse results from intra-modal parallel scanning. This design ensures that computations for each convolution kernel and input subregion remain independent, facilitating parallel processing of different convolution kernels or input blocks.
>
> **W2: Summation is too simple to model complex modality interaction**
>
> Our method utilizes the state transition matrix $S_m$ to evolve the dynamic process of multimodal fusion. While summing up historical states might seem straightforward, our model effectively filters historical information from different modalities through the coupling matrix $G$, retaining only the filtered, crucial information. Additionally, we introduce self-learning scalars that enable the model to adaptively learn the contribution of each modality to the task. Numerous experiments have demonstrated that, compared to other widely used fusion schemes such as averaging and cascading, our summation scheme not only enhances speed but also delivers superior performance.
>
> **W3: Comparison with other multi-modal Mamba models**
>
> Thank you for suggesting the concurrent multimodal Mamba work, including VL-Mamba and Cobra. Both VL-Mamba and Cobra are based on large language models (LLM) and are designed for VQA tasks. Our Coupled Mamba introduces Mamba to general multimodal tasks instead. In addition, VL-Mamba uses VSS Block to extract features from visual input, and then aligns textual features with visual features for fusion and inputs them into LMM for training. Cobra uses two visual models, DINOv2 and SigLIP, to extract visual features, and then aligns visual features with textual features for multimodal fusion. The Mamba models in both VL-Mamba and Cobra serve merely as feature extractors, and the multi-modal feature fusion is conducted within a LLM. They do not present any new schemes for multi-modal fusion. We will add both qualitative and quantitative comparisons with other concurrent multimodal Mamba works, including the proposed VL-Mamba and Cobra in the revision.
>
> **Q1: meaning of 'average (concat) fusion'**
>
> Yes, 'average fusion' directly averages multimodal features before processing them with a single Mamba, while 'concat fusion' concatenates multimodal features. We will make this point clearer.

---

> > ### Comment · Reviewer_nXDG · 2024-08-13
> >
> > Thanks for the response, especially for the part that clarified the process of parallelism. I've raised my score to borderline accept.

---

### Official Review · Reviewer_qhp2 · 2024-07-04

**Soundness:** 4
**Presentation:** 3
**Contribution:** 3
**Rating:** 6
**Confidence:** 4

**Summary:**

This paper proposes the coupled mamba to address the problem of multimodal data fusion. The core architecture of the coupled mamba is derived in the form of Equation (6) by improving upon Equation (5). Extensive ablation experiments validate the effectiveness of the proposed coupled mamba.

**Strengths:**

The article's logic is clear, the charts are coherent, and the proposed method alleviates the shortcomings of the original Mamba in handling multimodal tasks, contributing to the community.

**Weaknesses:**

1、I noticed that some of the references in the article are incomplete and inconsistent in format. Please handle this carefully. \
2、Why does $S_o$ in Algorithm 1 have the same form as in SSM? Is it reasonable to keep it consistent with SSM?\
3、The relationship between Equation (5) and the probability transition matrix in CHMH seems insignificant, although they are similar in form.\
4、The Equation (6) is a clever maneuver, but directly summing indicates that the coefficients in front of the hidden states of each modality are all 1. Is this design reasonable? Moreover, there is a lack of corresponding ablation experiments to validate this design.\
5、In Algorithm 1, the dimensions of $S_o$ : (B, L, E, N) and those of $S_m$ mentioned on page five are inconsistent.\
6、Some experiments, such as Table 1 and Table 2, lack corresponding inference time, FLOPs, or parameter data.\
7、In the Introduction, there is an extra space in 'Transformer-based'.

**Questions:**

1、Why is $S_o$ consistent with the form in SSM, rather than being obtained through input? What is its relationship with the probability transition matrix in Coupled Hidden Markov Models?\
2、What is the relationship between Formula 5 and the probability transition matrix in CHMH? Are they merely similar in form?\
3、In Formula 5, the term $\bar{A}_{m,m}$ should differ from other $\bar{A}$ terms, as it is most relevant to the current state $h_t^m$. Has this aspect been considered in this paper?\
4、Are there relevant ablation experiments to verify the validity of the simplification from Equation (5) to Equation (6)?

**Limitations:**

Please refer to Weaknesses and Limitations

---

> ### Author Rebuttal · Authors · 2024-08-07
>
> Thank you for your detailed and insightful comments and feedback, and we are deeply grateful for the time and expertise you dedicated to reviewing our paper..
>
> **W1: Incomplete and inconsistent references**
>
> We thank the reviewer for pointing out this issue. We will carefully proofread and correct all incorrect references, formatting issues, and typos.
>
> **W2: The form of $S_0$ is the same with SSM**
>
> In our Coupled Mamba, $S_0$ is obtained by multiplying the input-related $\overline{A}$ and the devised coupling matrix $G$. Please refer to the detailed proof in the appendix. It can be interpreted as a shared state transition matrix that transfers the coupling states based on a certain probability derived from the comprehensive state at time $t-1$. We use this same form for consistency with SSMs for unification and better interpretability.
>
> **W3: The relationship between Eqn. 5 and the probability transition matrix of CHMM**
>
> Thank you for the valuable feedback. CHMM has been widely adopted for multi-modal fusion [1, 2], utilizing the probability transition matrix to integrate information from multiple modalities. Our design of Equation 5 is inspired by CHMM, as the state propagation sequences in SSM resemble the state chains in CHMM. Equation 5 can also be interpreted as the probability transition between continuously evolving states, analogous to the transition of discrete states in CHMM.
>
> **W4: Eqn. 6 sets weights of hidden states from all modalities to 1**
>
> In our implementation, we use a learnable scalar to weight each modality in Eqn. 6. This has not been reflected in the paper, and we will clarify this point in the revision. Additionally, in response to your suggestion, we conducted a comparative experiment by setting the weights of all historical states to 1. The results are shown in Table 5 (in the PDF). By incorporating a learnable scalar to weight each modality in the equation, we address the issue of varying contributions from different modalities to the task, thereby improving the performance of multimodal fusion.
>
> **W5: Inconsistent dimensions of $s_0$ and $s_m$:**
> Thanks for pointing this issue out. Sorry for the typos. The correct dimensions for $S_{m}$ are [B,L,E,]. We will correct in the revision.
>
> **W6: Lack corresponding inference time, FLOPs, or parameter data in Table 1**
> We apologize for the missing information. In addition to providing only the inference time, FLOPs, or parameter data, we further compared our Coupled Mamba with several SOTA methods on inference time, FLOPS, and parameters. The results are shown in Fig. 2, Fig. 1, and Tab. 3 of the rebuttal PDF. Additionally, we also compared memory usage in Fig. 3. Our method exhibits the lowest memory consumption and fastest inference speed.
>
> **W7: In the Introduction, there is an extra space in 'Transformer-based**
>
> Thank you for your careful review. We will make revisions in future versions.
>
> **Q1**: Please refer to the reponse to **W2**.
>
> **Q2**: Please refer to the reponse to **W3**.
>
> **Q3:** $\bar{A}_{m,m}$ should differ from $\bar{A}$.
>
> Thank you for your suggestion and feedback. Note that all $\bar{A}{i,m}$ $, i \in 1 \ldots M$ in Eqn. 5 are learned parameters during network training. To preserve the generalizability of the model, we do not specifically emphasize the intra-modal transition $\bar{A}_{m,m}$, and instead let the network learn to prioritize modalities.
>
> **Q4: Ablation experiment on simplification from Eqn. 5 to Eqn. 6**
>
> Thank you for the valuable suggestion. We have conducted ablation study follow your suggestion, as illustrated in **Table 4** in the rebuttal PDF. The results indicate that this simplification reduces both the number of parameters and memory usage, and also improves inference speed, with minimal impact to the performance.
>
> [1]Garg A, Naphade M, Huang T S. Modeling video using input/output Markov models with application to multi-modal event detection[J]. Handbook of Video Databases: Design and Applications, 2003: 23-44.
>
> [2]ZHANG Yu-zhen,DING Si-jie,WANG Jian-yu,DAI Yue-wei,CHEN Qian.Event Detection by Fusing Multimodal Objects Using HMM[J].Journal of System Simulation,2012,24(8):1638-1642.

---

> > ### Comment · Reviewer_qhp2 · 2024-08-09
> >
> > Thanks for the responses. My concerns are resolved. I'll raise the score to WA.

---

### Official Review · Reviewer_FSJZ · 2024-07-11

**Soundness:** 3
**Presentation:** 3
**Contribution:** 3
**Rating:** 5
**Confidence:** 4

**Summary:**

The paper addresses the challenge of multi-modal fusion in deep learning. Current fusion methods struggle to capture complex intra- and inter-modality correlations. Recent state space models like Mamba show promise but are limited in fusing multiple modalities efficiently. The paper propose Coupled Mamba, key aspects include: A coupled state transition scheme that allows information exchange between modalities while maintaining individual propagation. A state summation and transition mechanism to enable parallel computing with multiple inputs. Derivation of a global convolution kernel for efficient parallelization.
The model architecture consists of multiple Coupled Mamba blocks, each processing different modalities and aggregating states before transitioning to the next time step.

Experiments were conducted on three multi-modal sentiment analysis datasets: CMU-MOSEI, CH-SIMS, and CH-SIMSV2.
Results show improved performance over existing methods as well as faster inference and GPU memory savings compared to transformer-based approaches. Effective performance on both aligned and unaligned multi-modal data.

**Strengths:**

1. The method demonstrates consistent performance and good efficiency compared with existing works.
2. The paper provides a detailed mathematical derivation of the coupled state transition process,
3. The idea of coupling mamba is novel.

**Weaknesses:**

The authors seem to have combined two Mamba models without providing enough insights into how this combination actually improves multi-modal fusion.

There's no clear analysis or interpretability of what the model is learning or how it's fusing information across modalities. This makes it difficult to understand why their approach is fundamentally better than existing methods, including transformers.

The speed and linear complexity benefits are inherited from the original mamba model, so they are apparent advantages for all state space-related works.

The experiments are limited to sentiment analysis tasks. The generalizability of the method to other multi-modal tasks or domains is not demonstrated or discussed.

**Questions:**

1. In table 9, what does Mamba Fusion mean?

2. Can the authors provide more interpretability behind combining to mamba structure and explain how the modalities interact between the two mamba?

**Limitations:**

The authors have addressed the limitations

---

> ### Author Rebuttal · Authors · 2024-08-07
>
> Thank you for your detailed and insightful comments and feedback, and we are deeply grateful for the time and expertise you dedicated to reviewing our paper..
>
> **W1: Provide insights over combining Mamba branches for multi-modal fusion**
>
> Effective multi-modal fusion hinges on balancing inter-modality information exchange while preserving independent intra-modality learning [1,2]. Our approach, Coupled Mamba, achieves this by integrating historical information across modalities while maintaining intra-modal state autonomy. Specifically, Coupled Mamba fuses the state of each modality's chain with adjacent chains from previous time steps across modalities. This integration ensures that each modality's current memory incorporates crucial historical context from multimodal data, progressively building a comprehensive model over time. We will make this point more clear.
>
> **W2: Lack analysis or interpretability of the model's multi-modal fusion mechanism**
>
> Thanks for the insightful feedback! In our Coupled Mamba, we use a separate Mamba branch for each modality, and fuses the state of each modality's chain with adjacent chains from previous time steps using a learnable state transition matrix ($A_{i, m}$ in Eqn. 5 and $S_{m}$ in Eqn. 6) to ehance multi-modality fusion. The state transition matrix selectively integrates important historical information from other modalities, while neglects information that's insignificant. By utilizing the coupling transition  between states, Coupled Mamba is tightly connected in time steps, effectively modeling multimodal data.
>
> This mechanism is fundamentally different from existing multimodal fusion schemes, where existing fusion methods can be categorized into
> 1. **attention mechanisms.** this method achieves multimodal information fusion through cross attention mechanisms.
> 2. **the encoder decoder method.** This method extracts high-level features of different modalities through an encoder, maps them to a low dimensional space, and then uses a decoder to generate predictions from these latent representations.
> 3. **Graph neural network method.** This method models multimodal data by constructing graph structured data.
>
> These designs all focus on learning good multi-modality features. In constrast, Coupled Mamba ensures intra-modal indenpence (as we use separate Mamba branches for each modality), and fuse important information from multi-modalities using the learnable state transition scheme, which emphasize more on the information fusion process. We believe that's the reason why our approach outperforms across all multi-modal tasks in experiments.
>
> We will explain this point clearer in the revision.
>
> **W3: The speed and linear complexity advantage is from Mamba**
>
> We only partially inherit the speed and linear complexity advantage of the original Mamba, by preserving the parallel scan scheme within each modality. However, the original Mamba and other state-space models were originally designed to process unimodal data and cannot handle cross-modal fusion well. When naively adapted for multi-modal data, as indicated by the 'concat' and 'mamba' fusion approach shown in Table 9 in the main paper, they require higher memory and computational resources, resulting in inferior performance compared to our coupled design.
>
> **W4: Experiments are limited to sentiment analysis tasks**
>
> Thanks for this helpful suggestion! We conducted additional experiments on the **BRCA benchmark** and **MM-IMDB benchmark**, and the results are presented in **Table 1,2** in our rebuttal PDF. The BRCA benchmark focuses on predicting the PAM50 subtype classification of breast cancer, based on three complex data modalities, i.e., the mRNA expression, DNA methylation and miRNA. while the MM-IMDB dataset is used for movie genre classification from movie intro, images. Despite the differences in data modalities and tasks, our Coupled Mamba achieves the best performance on both benchmarks compared to existing SOTA methods, demonstrating its generalizability.
>
> **Q1: Meaning of "mamba fusion" in Table 9**
>
> "mamba fusion" means establishing a seperate Mamba branch for each modality. Subsequently, the output features of these Mamba branches are merged through a weighted aggregation mechanism, consistent with the late-fusion paradigm embodied by methods such as LMF and Mult. We will further elaborate on this process in the revision.
>
>
> **Q2: Provide more interpretability of combining Mamba structures**:
>
> Please refer to our response to **W1**
>
> [1] Wang, Yikai, Wenbing Huang, Fuchun Sun, Tingyang Xu, Rong Yu, and Junzhou Huang. 2020. "Deep Multimodal Fusion by Channel Exchanging." NeurIPS 2020.\
> [2] Li, Yaowei, Ruijie Quan, Linchao Zhu, and Yi Yang. n.d. "Efficient Multimodal Fusion via Interactive Prompting." CVPR 2023

---

> > ### Comment · Reviewer_FSJZ · 2024-08-12
> >
> > Thanks for the responses. I am leaning towards accepting.

---

> > > ### Author Response · Authors · 2024-08-13
> > >
> > > Thank you for your positive feedback. We need to make a correction to our response to W3. While we do partially inherit the speed and linear complexity advantages of the original Mamba, our focus was on developing a fusion mechanism to extend Mamba's capability for better handling cross-modal data.
> > >
> > > When compared to naive adaptation methods like 'concat' and 'mamba fusion,' our Coupled Mamba has similar inference speed but consumes more memory due to the specialized mechanism we developed for cross-modality fusion. Despite the increased memory usage, the performance improvement from Coupled Mamba is relatively significant compared to these naive adaptations. Please refer to the 2nd-5th tables we presented to Reviewer JRDw for more details.

---

### Official Review · Reviewer_pbQ3 · 2024-07-11

**Soundness:** 3
**Presentation:** 3
**Contribution:** 3
**Rating:** 6
**Confidence:** 3

**Summary:**

This work proposes Coupled SSM (State Space Models) to fuse multiple modalities effectively with SSM. Instead of fusing multi-modal features directly, the proposed method couples state chains of multiple modalities while maintaining the independence of intra-modality state processes. Specifically, they first propose an inter-modal hidden states transition scheme to fuse multiple modalities effectively. Then, they propose an expedited coupled state transition scheme to adapt the hardware-aware parallelism of SSMs for efficiency.  Experimental results on classification task on CMU-MOSEI, CH-SIMS, CH-SIMSV2 show promising performance of the proposed method.

**Strengths:**

1. The proposed method couples the state chains of multiple modalities while maintaining the independence of intra-modality state processes.
2. The proposed method is more memory efficient compared to cross-attention with the increasing sequence length.
3. Results on several benchmark datasets show the superior performance of the proposed method.

**Weaknesses:**

1. The improvement in the regression task on CMU-MOSEI seems marginal compared to the baselines.
2. In line 201, the authors conclude that the results in Tables 3 and 4 show the SOTA performance of the proposed method regardless of whether the data are aligned or not. However, both results in Tables 3 and 4 are performed on unaligned data. Besides, the robustness of the proposed method needs further verified, since I cannot find the results that support the claims about robustness.

**Questions:**

Please refer to the weaknesses.

**Limitations:**

The discussions about the limitations of this work would be further improved.

---

> ### Author Rebuttal · Authors · 2024-08-07
>
> Thank you for your detailed and insightful comments and feedback, and we are deeply grateful for the time and expertise you dedicated to reviewing our paper..
>
> **W1: The improvement in the regression task on CMU-MOSEI seems marginal**
>
> The CMU-MOSEI dataset mostly consists of short sequences: most of the videos, audios, and texts are only 1-5 seconds in duration. Mamba is more suitable for modeling relatively long sequences, which results in marginal improvement of Coupled Mamba on the CMU-MOSEI dataset. In contrast, for datasets with longer sequences, such as data in CH-SIMS, CH-SIMSV2, BRCA, and MM-IMDB datasets, our Coupled Mamba can effectively enhance multi-modal fusion and outperforms the state-of-the-arts by a relatively large margin. We will include a discussion on the impact of sequence lengths in the limitations section.
>
> **W2: Less experiments on aligned data and evidences of robustness:**
>
> The fewer experiments on 'aligned' data are influenced by the composition of the datasets used, where there is a predominance of unaligned data. This is reasonable, as aligned data is more difficult to collect. Moreover, many existing classification methods for the CMU-MOSEI dataset primarily focus on unaligned data; hence, we use unaligned data for a fair comparison. Besides, experiments on aligned data in the main paper Table 1 have demonstrated our excellent performance for aligned data input. Furthermore, unaligned data is more challenging to process, and our approach achieves superior performance regardless of aligned or unaligned data.
>
> For robustness, we have conducted experiments on aligned/unaligned data (Table 1 of the main paper), various modalities (text, audio, video, mRNA, miRNA, DNA, as shown in Tables 1 and 2 in the rebuttal PDF), sequence lengths (ranging from 100 to 500), and text in different language domains (English vs. Chinese). All experiments show that our Coupled Mamba achieves superior performance, demonstrating its robustness across various data input.

---

> > ### Comment · Reviewer_pbQ3 · 2024-08-10
> >
> > First, I mean the claim "It can be seen from the results...the proposed fusion method achieves state-of-the-art (SOTA) regardless of whether the data are aligned or not" is not rigorous, as experiments in Tables 3,  4, and 10 are conducted on unaligned data. It would be better to consider the results from Table 1 and make such a conclusion.
> >
> > Second, I think the author may have confused robustness with generalization, and I believe robustness is about the model performance under noisy inputs.

---

> > > ### Author Response · Authors · 2024-08-12
> > >
> > > Thank you for your further feedback.
> > >
> > > **On the unrigorous conclusions from Tables 3, 4, and 10**: We apologize for the misunderstanding. You are correct that the expression was unrigorous if it referred only to Tables 3, 4, and 10. In the revision, we will update the phrase to 'It can be seen from Tables 1, 3, 4, and 10 ...' for a more accurate and precise expression.
> > >
> > > **On the evaluation of robustness**: We apologize for the confusion. To validate the robustness of our model, we tested our Coupled Mamba under conditions where part of the data was missing, and with noisy input (by adding a certain level of Gaussian noise).
> > >
> > > For testing on missing data, we followed the experimental setup in [1]. We conducted experiments on the CMU-MOSEI dataset by creating a random mask with the same shape as the original tensor, where each element is drawn from the Bernoulli distribution B(1-p). This means each element has a p% probability of being 1 (retained) and a (1-p)% probability (i.e., the missing rate (MR)) of being 0 (missing). We then multiplied this random mask with the original tensor, causing the areas with a mask value of 0 to result in missing data in the original tensor. The results are presented in the table below, with the numbers for other baselines copied from [1] due to rebuttal time constraints. Our method demonstrates the best performance. Please note that the left side of **/** shows Acc_2, while the right side denotes the F1 Score.
> > >
> > > | Datasets   | MR | DCCA   | DCCAE  | MCTN   | MMIN   | GCNET  | **Coupled Mamba** |
> > > |------------|----|--------|--------|--------|--------|--------|------------------|
> > > | CMU-MOSEI  | 0.0 | 80.7/80.9 | 81.2/81.2 | 84.2/84.2 | 84.3/84.2 | 85.2/85.1 | **85.5/85.6**    |
> > > |            | 0.1 | 77.4/77.3 | 78.4/78.3 | 81.8/81.6 | 81.9/81.3 | 82.3/82.1 | **82.6/82.7**    |
> > > |            | 0.2 | 73.8/74.0 | 75.5/75.4 | 79.0/78.7 | 79.8/78.8 | 80.3/79.9 | **81.1/80.9**    |
> > > |            | 0.3 | 71.1/71.2 | 72.3/72.2 | 76.9/76.2 | 77.2/75.5 | 77.5/76.8 | **81.0/81.0**    |
> > > |            | 0.4 | 69.5/69.4 | 70.3/70.0 | 74.3/74.1 | 75.2/72.6 | 76.0/74.9 | **78.4/78.5**    |
> > > |            | 0.5 | 67.5/65.4 | 69.2/66.4 | 73.6/72.6 | 73.9/70.7 | 74.9/73.2 | **77.4/77.7**    |
> > > |            | 0.6 | 66.2/63.1 | 67.6/63.2 | 73.2/71.1 | 73.2/70.3 | 74.1/72.1 | **75.1/75.4**    |
> > > |            | 0.7 | 65.6/61.0 | 66.6/62.6 | 72.7/70.5 | 73.1/69.5 | 73.2/70.4 | **74.1/74.2**    |
> > > | Average    |     | 70.3/71.2 | 72.6/71.2 | 77.0/76.1 | 77.3/75.4 | 77.9/76.8 | **79.4/79.5**    |
> > >
> > > For the evaluation on noisy input, we added Gaussian noise to the input at three noise levels, with standard deviations of 1, 2, and 3. Due to the limited time available for the rebuttal, we were only able to compare our results with Mult [2] as a reference. The performance of Coupled Mamba declines much slower as the noise level increases. The left and right sides of **/** represent Acc_2 and F1 Score, respectively.
> > >
> > > |                       | Std=0            | Std=1            | Std=2            | Std=3            |
> > > |-----------------------|------------------|------------------|------------------|------------------|
> > > | Mult                  | 82.5 / 82.3      | 79.8 / 80.1      | 77.1 / 76.9      | 74.6 / 74.8      |
> > > | CrossAttention       | 84.6 / 84.5      | 82.3 / 82.2      | 80.2 / 80.4      | 78.4 / 78.6      |
> > > | **Coupled Mamba**     | **85.6 / 85.5** | **84.6 / 84.3** | **83.2 / 83.3** | **81.4 / 81.3** |
> > >
> > > The experiments on missing data and noisy input demonstrate that our Coupled Mamba model is robust.
> > >
> > > **DCCA** Andrew G, Arora R, Bilmes J, et al. Deep canonical correlation analysis[C]//International conference on machine learning. PMLR, 2013: 1247-1255.
> > >
> > > **DCCAE** Wang W, Arora R, Livescu K, et al. On deep multi-view representation learning[C]//International conference on machine learning. PMLR, 2015: 1083-1092.
> > >
> > > **MCTN** Pham H, Liang P P, Manzini T, et al. Found in translation: Learning robust joint representations by cyclic translations between modalities[C]//Proceedings of the AAAI conference on artificial intelligence. 2019, 33(01): 6892-6899.
> > >
> > > **MMIN** Zhao J, Li R, Jin Q. Missing modality imagination network for emotion recognition with uncertain missing modalities[C]//Proceedings of the 59th Annual Meeting of the Association for Computational Linguistics and the 11th International Joint Conference on Natural Language Processing (Volume 1: Long Papers). 2021: 2608-2618.
> > >
> > > **GCNET** Lian Z, Chen L, Sun L, et al. GCNet: Graph completion network for incomplete multimodal learning in conversation[J]. IEEE Transactions on pattern analysis and machine intelligence, 2023, 45(7): 8419-8432.
> > >
> > > [1] Wang Y, Li Y, Cui Z. Incomplete multimodality-diffused emotion recognition[J]. NeurIPS,2024,36.
> > >
> > > [2] Tsai Y H H, Bai S, Liang P P, et al. Multimodal transformer for unaligned multimodal language sequences[C],ACL 2019.

---

### Official Review · Reviewer_JRDw · 2024-07-14

**Soundness:** 2
**Presentation:** 2
**Contribution:** 3
**Rating:** 6
**Confidence:** 4

**Summary:**

The paper introduces a coupled mamba model for multi-modal fusion. The multi-modal hidden states are fused inside of Mamba, so the current state learns not only from a single modality but the correlation of all modalities. The experiments on multi-modal sentiment analysis show that the proposed model outperforms other baselines.

**Strengths:**

- The proposed method makes sense, and it outperforms other baselines.
- The motivation is reasonable.
-  The ablation study shows that the coupled fusion is better than other common fusion approaches (cross-attention, average, concat, and simple fusion).

**Weaknesses:**

1. Lack of speed and memory analysis: An important aspect of the proposed method is the memory and speed overhead. It's expected that the overhead is not more than cross-attention (as shown in Figures 3 and 4). However, the paper missed comparisons with other fusion methods, especially the ones that use the inputs directly instead of hidden states.

2. The experiments are only on multi-modal sentiment analysis. I believe the proposed method is pretty general for any multi-modal data and tasks. Showing a variety of applications will make the paper more convincing and appealing.

**Questions:**

1. The original Mamba uses a parallel scan (recurrent representation) for parallel computing. However, the authors converted Mamba to the convolutional representation. Since eq 6 mainly adds the addition operation (to combine hidden states from different modalities), it should work with the parallel scan. What is the reason to convert to the convolutional representation for parallelization?
2. The proposed method is compared with average, concat, and mamba fusions in the ablation study, but the explanation of these methods is missing. I assume the average and concat are done directly on the input. How does the mamba fusion work?
3. What does unaligned/aligned mean in the experiments (Tables 1 and 6)?
4. Some details about the dataset are missing:
    - What's the sequence length of each dataset?
    - What modalities do the datasets contain?

**Limitations:**

The limitation is discussed as a part of the ethical implications discussion.

---

> ### Author Rebuttal · Authors · 2024-08-07
>
> Thank you for your thorough and insightful feedback. We genuinely appreciate the effort and expertise you invested in reviewing our paper.
>
> **W1: Lack of speed and memory analysis**
>
> We thank the reviewer for this insightful feedback. In our rebuttal PDF, we have included a comparison of memory usage(**Fig 3**), inference speed(**Fig 2**), flops(**Fig 1**), and parameter size (**Table 3**) with Mult[1], MISA[2], TFN[3], and LMF[4], all using inputs directly instead of hidden states. Additionally, Tab. 3 presents a detailed comparison of parameters. From both Fig. 1 and Tab. 3, it is evident that our Coupled Mamba consumes significantly less memory and achieves the fastest inference speed for sequences from short to long.
>
> **W2: Expriments on more multi-modal data and tasks to demonstrate generalizability**
>
> Thanks for this helpful suggestion! We conducted additional experiments on the **BRCA benchmark** and **MM-IMDB benchmark**, and the results are presented in **Tab 1,2** in our rebuttal PDF. The BRCA benchmark focuses on predicting the PAM50 subtype classification of breast cancer from mRNA, DNA, miRNA data, while the MM-IMDB dataset is used for movie genre classification from text and images. Despite the differences in data modalities and tasks, our Coupled Mamba achieves the best performance on both benchmarks compared to existing SOTA methods, demonstrating its generalizability.
>
> **Q1: Reason of converting to the convolutional representation for parallelization**
>
> The parallel scanning scheme in Mamba computes independent partial states recurrently from inputs and then combines these partial states using a global convolution in a hierarchical manner to achieve parallelism. In our Coupled Mamba, we adapt this approach by conducting recurrently computation of partial states within each modality. We enhance this process with a summation operation through a global convolutional kernel, enabling multi-modal fusion capabilities while preserving the inherent advantages of parallelism. To be more specific, the parallel scanning process generates intermediate results, which we fuse using a summation scheme with a global convolutional kernel across modalities. This scheme is inherently parallelizable and accelerates the fusion of multimodal data. We will clarify this point further in the revision.
>
> **Q2: Explanation of *average*, *concat*, and *mamba* fusions in the ablation study**
>
> **average** fusion involves directly averaging the multi-modal features. **concat** fusion concatenates the multi-modal features. The fused feature is then fed into a Mamba model, followed by a pooling and head layer for the final output. For **Mamba fusion**, we create a separate Mamba branch for each modality. The output features of each branch are then combined using a weighted summation (following late fusion schemes as in LMF, Mult). We will clearify this in revision.
>
> **Q3: The meaning of *unaligned* and *aligned* in experiments**
>
> **aligned** data refers to data from different modalities that are precisely synchronized in both spatial and temporal dimensions, such as synchronized audio and video. In contrast, **unaligned** data refers to instances where different modalities are not synchronized, presenting a more challenging scenario. Our method demonstrates superior performance on both aligned and unaligned data.
>
> **Q4: Some details about the dataset are missing**
>
> Our experiments utilize datasets comprising data from **video, audio, and text modalities**. The CMU-MOSEI dataset includes both aligned and unaligned data, with sequence lengths of 50 for aligned text, audio, and video, and 50, 500, and 375 for unaligned text, audio, and video, respectively. The CH-SIMS and CH-SIMSV2 datasets contain exclusively unaligned data, with sequence lengths of 39, 400, and 55 for text, audio, and video in CH-SIMS, and 50, 925, and 232 in CH-SIMSV2.
>
> [1] Tsai, Yao-Hung Hubert, Shaojie Bai, Paul Pu Liang, J. Zico Kolter, Louis-Philippe Morency, and Ruslan Salakhutdinov. 2019. “Multimodal Transformer for Unaligned Multimodal Language Sequences.” In Proceedings of the 57th Annual Meeting of the Association for Computational Linguistics. doi:10.18653/v1/p19-1656.
>
> [2] Hazarika, Devamanyu, Roger Zimmermann, and Soujanya Poria. 2020. “MISA: Modality-Invariant and -Specific Representations for Multimodal Sentiment Analysis.” Cornell University - arXiv,Cornell University - arXiv, May.
>
> [3] Zadeh, Amir, Minghai Chen, Soujanya Poria, Erik Cambria, and Louis-Philippe Morency. 2017. “Tensor Fusion Network for Multimodal Sentiment Analysis.” arXiv: Computation and Language,arXiv: Computation and Language, July.
>
> [4] Liu, Zhun, Ying Shen, Varun Bharadhwaj Lakshminarasimhan, Paul Pu Liang, AmirAli Bagher Zadeh, and Louis-Philippe Morency. 2018. “Efficient Low-Rank Multimodal Fusion with Modality-Specific Factors.” In Proceedings of the 56th Annual Meeting of the Association for Computational Linguistics (Volume 1: Long Papers). doi:10.18653/v1/p18-1209.

---

> > ### Comment · Reviewer_JRDw · 2024-08-12
> > **response to the rebuttal**
> >
> > Thank you for the answers.
> > I still have a concern regarding speed and memory analysis.
> > The authors added Figures 1-3 in the rebuttal pdf (FLOPS, inference speed, and memory comparisons). However, the comparing methods are not SSM/Mamba-based, and none of these models are recent SOTA. It's unclear whether the memory and speed benefits come from Mamba. Could authors add these additional models to Figures 1-3?
> >
> > 1) different fusion methods (avg. concat, and mamba fusion from Table 9)
> > 2) more recent methods like [44]
> > 3) some models from Tables 1 and 2 (rebuttal pdf)
> >
> > Also, please add citations for Tables 1 and 2 in the rebuttal pdf. It's unclear whether these models are comparable.

---

> > > ### Author Response · Authors · 2024-08-13
> > >
> > > Thank you for your thoughtful feedback. Follow your suggestion, we conducted additional experiments on memory and inference speed using five methods: Average, Concat, Mamba Fusion, IMDer, and DynMM. We present the result in the below table.
> > >
> > > |              | Seq=100 | Seq=200 | Seq=300 | Seq=400 | Seq=500 |
> > > |--------------|----------|----------|----------|----------|----------|
> > > | **Average**      | 24.58/0.0006 | 41.98/0.0006 | 59.39/0.0006 | 76.80/0.0006 | 94.20/0.0006 |
> > > | **Concat**       | 49.15/0.0006 | 95.23/0.0006 | 131.04/0.0007   | 174.08/0.0007 | 205.84/0.0007 |
> > > | **Mamba Fusion** | 41.98/0.0068  | 60.42/0.0068  | 80.90/0.0070 | 103.42/0.0071  | 134.57/0.0073 |
> > > | **IMDer**        | 464.37/0.7154   | 492.74/0.7352  | 535.41/0.7587  | 650.87/0.7994  | 693.54/0.8124  |
> > > | **DynMM**        | 62.46/0.2821  | 226.30/0.3133 | 301.06/0.3314 | 353.28/0.3546 | 467.97/0.3915 |
> > > | **Coupled Mamba**| 58.36/0.0069  | 103.42/0.0071 | 151.00/0.0073   | 205.82/0.0074 | 244.73/0.0075 |
> > >
> > > The data on the left and right sides of the **/** represent memory usage (MB) and inference speed (s), respectively. The results show that Coupled Mamba is significantly faster (by 50 to 100 times), and consuming less memory than both IMDer and DynMM. Moreover, our Coupled Mamba outperforms Mamba Fusion (Table 9 in main paper) with similar speed and slightly higher memory usage, demonstrating that our design is both efficient and effective.
> > >
> > > **Here are some articles that supplement the baseline comparisons in the experiments:**
> > >
> > > **CRidge** : Van De Wiel M A, Lien T G, Verlaat W, et al. Better prediction by use of co‐data: adaptive group‐regularized ridge regression[J]. Statistics in medicine, 2016, 35(3): 368-381.
> > >
> > > **GMU** : Arevalo J, Solorio T, Montes-y-Gómez M, et al. Gated multimodal units for information fusion[J]. arXiv preprint arXiv:1702.01992, 2017.
> > >
> > > **CF**: Hong D, Gao L, Yokoya N, et al. More diverse means better: Multimodal deep learning meets remote-sensing imagery classification[J]. IEEE Transactions on Geoscience and Remote Sensing, 2020, 59(5): 4340-4354.
> > >
> > > **MOGONET**: Wang T, Shao W, Huang Z, et al. MOGONET integrates multi-omics data using graph convolutional networks allowing patient classification and biomarker identification[J]. Nature communications, 2021, 12(1): 3445.
> > >
> > > **TMC** : Han Z, Zhang C, Fu H, et al. Trusted multi-view classification[C]//International Conference on Learning Representations. 2020.
> > >
> > > **MM-Dynamics** : Han Z, Yang F, Huang J, et al. Multimodal dynamics: Dynamical fusion for trustworthy multimodal classification[C]//Proceedings of the IEEE/CVF conference on computer vision and pattern recognition. 2022: 20707-20717.
> > >
> > > **LRMF**: Liu Z, Shen Y, Lakshminarasimhan V B, et al. Efficient low-rank multimodal fusion with modality-specific factors[J]. arXiv preprint arXiv:1806.00064, 2018.
> > >
> > > **MFM** : Tsai Y H H, Liang P P, Zadeh A, et al. Learning factorized multimodal representations[J]. arXiv preprint arXiv:1806.06176, 2018.
> > >
> > > **MI-Matrix** : Jayakumar S M, Czarnecki W M, Menick J, et al. Multiplicative interactions and where to find them[C]//International conference on learning representations. 2020.
> > >
> > > **RMFE** : Gat I, Schwartz I, Schwing A, et al. Removing bias in multi-modal classifiers: Regularization by maximizing functional entropies[J]. Advances in Neural Information Processing Systems, 2020, 33: 3197-3208.
> > >
> > > **CCA**: Sun Z, Sarma P, Sethares W, et al. Learning relationships between text, audio, and video via deep canonical correlation for multimodal language analysis[C]//Proceedings of the AAAI conference on artificial intelligence. 2020, 34(05): 8992-8999.
> > >
> > > **RefNet**: Sankaran S, Yang D, Lim S N. Multimodal fusion refiner networks[J]. arXiv preprint arXiv:2104.03435, 2021.
> > >
> > > **DynMM** : Xue Z, Marculescu R. Dynamic multimodal fusion[C]//Proceedings of the IEEE/CVF Conference on Computer Vision and Pattern Recognition. 2023: 2575-2584.

---

> > > > ### Comment · Reviewer_JRDw · 2024-08-13
> > > >
> > > > Thank you for the additional table.
> > > >
> > > > I understand that coupled Mamba outperforms other naive fusion approaches, but the proposed appraoch requires almost 2x more memory (and similar inference speed when the sequence length=500) compared to mamba fusion while the accuracy improvement is minimal.
> > > >
> > > > The paper does not clearly indicate that the efficiency of the approach is by using Mamba for multi-modal fusion, not by introducing coupled Mamba. The response to reviewer FSJZ10 also indicates that the coupled design is more efficient than other fusion approaches.
> > > > > When naively adapted for multi-modal data, as indicated by the 'concat' and 'mamba' fusion approach shown in Table 9 in the main paper, they require higher memory and computational resources, resulting in inferior performance compared to our coupled design.
> > > >
> > > > The authors should clarify this point in the paper and to other reviewers as it can be misleading.

---

> > > > > ### Author Response · Authors · 2024-08-13
> > > > >
> > > > > Thank you for your thoughtful comments. We acknowledge that Coupled Mamba consumes more memory with similar inference speed. This is due to the specialized mechanism we developed for cross-modality fusion, which increases memory usage.
> > > > >
> > > > > Regarding the minimal accuracy improvement, this is because our comparison was based on the CMU-MOSEI dataset (Table 9 in our paper), which involves relatively short sequences. Our approach is more suitable for modeling longer sequences. (Reviewer pbQ3 also mentioned this issue in W1; please also check our response.) To provide a more comprehensive understanding, we present memory, speed, and accuracy results in the table below for all other datasets used in the paper, i.e., the CH-SIMS, CH-SIMSV2, BRCA, and MM-IMDB datasets. It can be seen that the improvement from Coupled Mamba is not marginal compared to Mamba Fusion for datasets with longer sequences.
> > > > >
> > > > > |          | **CH-SIMS**       |        |               |
> > > > > |----------|---------------|---------------|---------------|
> > > > > | **Model** | **Acc_2**     | **F1-Score**  |**Infer(s)/Mem(MB)**|
> > > > > | Mamba Fusion | 80.1          | 79.7          |0.0070/93.18|
> > > > > | **Coupled Mamba** | **81.8** | **81.3** |0.0072/177.15|
> > > > >
> > > > > |      | **CH-SIMSV2**      |      |               |
> > > > > |------|----------------|----------------|---------------|
> > > > > | **Model** | **Acc_2** | **F1-Score** |**Infer(s)/Mem(MB)**|
> > > > > | Mamba Fusion | 81.5          | 81.7          |0.0071/89.01|
> > > > > | **Coupled Mamba** | **83.4** | **83.5** |0.0074/171.01|
> > > > >
> > > > > | **BRCA** | **Acc**      | **WeightedF1** | **MacroF1**    |**Infer(s)/Mem(MB)**|
> > > > > |------|----------|------------|------------|---------------|
> > > > > | Mamba Fusion           | 84.7      | 84.9      | 81.7      |0.0067/67.59|
> > > > > | **Coupled Mamba** | **88.1**  | **88.5**  | **85.4**  |0.0069/126.97|
> > > > >
> > > > > | **MM-IMDB** | **MicroF1** | **MacroF1** |**Infer(s)/Mem(MB)**|
> > > > > |-------------|-------------|-------------|-------------|
> > > > > | Mamba Fusion | 60.27       | 49.84       |0.051/171.01|
> > > > > | **Coupled Mamba** | **62.41** | **52.58** |0.052/343.04|
> > > > >
> > > > > We also apologize for the misleading conclusion regarding memory and speed on 'concat' and 'Mamba Fusion.' We will clarify this point to Reviewer FSJZ and make the necessary corrections in the paper.

---

> > > > > > ### Comment · Reviewer_JRDw · 2024-08-14
> > > > > >
> > > > > > Thank you for the additional experiments. This is very helpful.
> > > > > > As my last concern is addressed, I raise my rating from 5 to 6.
> > > > > >
> > > > > > Please add the answers and additional results to the revised version. Also, the conclusion of the last discussion regarding efficiency should be clarified in the paper. It's important to note that the proposed approach is not necessarily efficient compared to mamba-based fusion models but improves performance.

---

> > > > > > > ### Author Response · Authors · 2024-08-14
> > > > > > >
> > > > > > > Thank you very much for the supportive feedback. We will meticulously revise the paper and clarify the points concerning efficiency and performance.

---

### Author Rebuttal · Authors · 2024-08-07

We thank the reviewers for their insightful feedback and appreciate the time and expertise they have dedicated to evaluating our work. We are encouraged by the positive comments highlighting the strengths of our approach: "superior and consistent performance" (JRDw, pbQ3, FSJZ, qhp2, nXDG), "demonstrates good efficiency" (pbQ3, FSJZ, nXDG), "clear logic & motivating direction" (JRDw, qhp2, nXDG), "novel idea" (FSJZ), and "detailed mathematical derivation" (FSJZ, nXDG).


Reviewers expressed concerns regarding the lack of speed and memory analysis (JRDw, FSJZ, qhp2), and the need for experiments on other multi-modal tasks besides sentiment analysis (JRDw, FSJZ), lack ablation on Eqn. 5 and Eqn. 6 (qhp2). In response, we have conducted additional experiments and included detailed figures and tables in the rebuttal PDF to address these concerns, coupled with our reviewer-specific replies.


**Summary of figures and tables in the rebuttal PDF in response to reviewers' feedback:**

**Table 1**: Experimental results on the **BRCA benchmark**, classifying breast cancer PAM50 using mRNA (mR), DNA methylation (D), and miRNA (miR) expression data.

**Table 2**: Experimental results on the **MM-IMDB Benchmark**, classifying movie categories using image (I) and text (T) modalities. Our method achieved the best results in both MicroF1 and MacroF1 indicators, with a **2.06%** improvement in MicroF1.


**Table 3**:  Comparison of parameters across different methods under identical conditions (same number of layers, hidden dimensions, etc.), demonstrating Coupled Mamba's superior performance with reduced complexity.

**Figure1, 2, and 3**: Showing FLOPs, Inference Time, and Memory Usage across varying sequence lengths for different models. Coupled Mamba exhibits lower computing resource requirements, faster inference speed, and significantly reduced memory usage.

**Table 4**: Comparison between Eqn. 5 and its simplified version Eqn. 6, showing effective reduction in model parameters, inference time, and memory usage with our simplification.

**Table 5**: Results comparing fixed weights (all modalities set to 1) and adaptive scalar learning weights in Eqn. 6 on the CH-SIMS dataset. Adaptive learning scalars improve model performance by enabling nuanced multimodal fusion, learning varying contributions of modalities to the task.

**Here are some articles that supplement the baseline comparisons in the experiments:**\
**CRidge** : Van De Wiel M A, Lien T G, Verlaat W, et al. Better prediction by use of co‐data: adaptive group‐regularized ridge regression[J]. Statistics in medicine, 2016, 35(3): 368-381.\
**GMU** : Arevalo J, Solorio T, Montes-y-Gómez M, et al. Gated multimodal units for information fusion[J]. arXiv preprint arXiv:1702.01992, 2017.\
**CF**: Hong D, Gao L, Yokoya N, et al. More diverse means better: Multimodal deep learning meets remote-sensing imagery classification[J]. IEEE Transactions on Geoscience and Remote Sensing, 2020, 59(5): 4340-4354.\
**MOGONET**: Wang T, Shao W, Huang Z, et al. MOGONET integrates multi-omics data using graph convolutional networks allowing patient classification and biomarker identification[J]. Nature communications, 2021, 12(1): 3445.\
**TMC** : Han Z, Zhang C, Fu H, et al. Trusted multi-view classification[C]//International Conference on Learning Representations. 2020.\
**MM-Dynamics** : Han Z, Yang F, Huang J, et al. Multimodal dynamics: Dynamical fusion for trustworthy multimodal classification[C]//Proceedings of the IEEE/CVF conference on computer vision and pattern recognition. 2022: 20707-20717.\
**LRMF** : Liu Z, Shen Y, Lakshminarasimhan V B, et al. Efficient low-rank multimodal fusion with modality-specific factors[J]. arXiv preprint arXiv:1806.00064, 2018.\
**MFM** : Tsai Y H H, Liang P P, Zadeh A, et al. Learning factorized multimodal representations[J]. arXiv preprint arXiv:1806.06176, 2018.\
**MI-Matrix** : Jayakumar S M, Czarnecki W M, Menick J, et al. Multiplicative interactions and where to find them[C]//International conference on learning representations. 2020.\
**RMFE** : Gat I, Schwartz I, Schwing A, et al. Removing bias in multi-modal classifiers: Regularization by maximizing functional entropies[J]. Advances in Neural Information Processing Systems, 2020, 33: 3197-3208.\
**CCA** : Sun Z, Sarma P, Sethares W, et al. Learning relationships between text, audio, and video via deep canonical correlation for multimodal language analysis[C]//Proceedings of the AAAI conference on artificial intelligence. 2020, 34(05): 8992-8999.\
**RefNet** : Sankaran S, Yang D, Lim S N. Multimodal fusion refiner networks[J]. arXiv preprint arXiv:2104.03435, 2021.\
**DynMM** : Xue Z, Marculescu R. Dynamic multimodal fusion[C]//Proceedings of the IEEE/CVF Conference on Computer Vision and Pattern Recognition. 2023: 2575-2584.

---

### Decision · Program_Chairs · 2024-09-25

**Decision:**

Accept (poster)

**Comment:**

This work proposes Coupled SSM (State Space Models) to fuse multiple modalities effectively with SSM. Instead of fusing multi-modal features directly, the proposed method couples state chains of multiple modalities while maintaining the independence of intra-modality state processes. Specifically, they first propose an inter-modal hidden states transition scheme to fuse multiple modalities effectively. Then, they propose an expedited coupled state transition scheme to adapt the hardware-aware parallelism of SSMs for efficiency. Experimental results on classification task on CMU-MOSEI, CH-SIMS, CH-SIMSV2 show promising performance of the proposed method.